# Trends in HIV Testing and Associated Factors among Adolescent Girls and Young Women in Zimbabwe: Cross-Sectional Analysis of Demographic and Health Survey Data from 2005 to 2015

**DOI:** 10.3390/ijerph19095165

**Published:** 2022-04-24

**Authors:** Abgail Pachena, Alfred Musekiwa

**Affiliations:** School of Health Systems and Public Health, Faculty of Health Sciences, University of Pretoria, Pretoria 0002, South Africa; alfred.musekiwa@up.ac.za

**Keywords:** HIV testing, adolescent girls and young women, trends, associated factors, sub-Saharan Africa, Zimbabwe

## Abstract

Adolescent girls and young women (AGYW), aged 15–24 years, experience higher HIV incidence compared to their male counterparts, especially in sub-Saharan Africa (SSA), where the HIV burden is highest. This study determined trends in self-reported HIV testing and associated factors among AGYW in Zimbabwe using the Zimbabwe DHS datasets for 2005/6, 2010/11, and 2015. The proportion of adolescents aged 15–19 years who had ever tested for HIV increased from 14.7% in 2005/6 to 26.5% in 2010/11 and 47.9% in 2015. Among young women, aged 20–24 years, the proportion increased from 34.8% in 2005/6 to 68.7% in 2010/11 and 84.8% in 2015. The odds of ever having an HIV test were significantly higher for those with a higher education (adjusted odds ratio [aOR] 12.49, 95% CI: 2.69 to 57.92, *p* = 0.001), comprehensive HIV knowledge (aOR 1.91, 95% CI: 1.31 to 2.78, *p* = 0.001), knowledge about mother-to-child transmission (MTCT) (aOR 2.09, 95% CI: 1.55 to 2.82, *p* < 0.001), non-discriminatory attitudes (aOR 1.60, 95% CI: 1.12 to 2.28, *p* = 0.010), three or more lifetime sexual partners (aOR 2.0, 95% CI: 1.09 to 3.66, *p* = 0.025), and a history of pregnancy (aOR 6.08, 95% CI: 4.22 to 8.75, *p* < 0.001). There is need to scale-up programmes targeting AGYW.

## 1. Introduction

According to the World Health Organization (WHO), HIV infection had caused 33 million deaths by the end of 2020 globally since the start of the pandemic. In 2019, 38 million people globally were living with HIV. Africa accounted for two-thirds of these infections [1]. Thus, HIV infection is of public health importance because of the high burden of disease.

The Joint United Nations Programme on HIV/AIDS (UNAIDS) had 90–90–90 global targets set for 2020 to reduce HIV disease burden and of ultimately having no new HIV infections by 2030 through achieving 95–95–95 targets. These targets mean that “95% of all those living with HIV are aware of their status, 95% of those with a positive HIV diagnosis are on treatment, and 95% of those on treatment achieve sustained viral suppression” [2]. Although there has been progress in reducing new HIV infections and deaths attributable to HIV, the 90–90–90 UNAIDS 2020 targets were missed, with global achievements currently around 81–67–59 [1].

Key populations accounted for 62% of new HIV infections by 2020 [1]. Key populations include men who have sex with men (MSM), people in prisons and other closed spaces, transgender people, those who inject drugs, and sex workers, of which the majority are women, including adolescent girls and young women (AGYW), aged 15–24 years [1,3]. The risk of acquiring HIV infection among sex workers is 26-times higher than that of the general population [3].

Globally, HIV-related illnesses are the leading cause of mortality among women [1]. The incidence of HIV infection among AGYW is quite high, with 340,000 new HIV infections having occurred in 2017 [4]. Globally, there were 5000 new HIV infections per week among AGYW in 2020 [3]. This group is disproportionately affected by HIV infection more than males in the same age group [4,5,6]. This is mainly due to gender inequality and cultural norms that do not support women empowerment [4]. Age-disparate relationships significantly contribute to new HIV infections among AGYW [7,8]. Globally, three out of every five new HIV infections that occur in young people are in females [4]. Sub-Saharan Africa (SSA) has the highest HIV burden of disease, including among AGYW [3,4,6]. Six out of every seven new HIV infections among adolescents occur in girls [3]. In SSA, there were 4200 new HIV infections per week among AGYW in 2020 [3]. Globally, 90% of HIV deaths among AGYW occurred in SSA in 2017 [4]. Although AGYW make up 10% of the population in SSA, 59% of new HIV infections occurred in this group in 2019 [4].

Zimbabwe is one of the countries in SSA that has a high burden of HIV infection, with an incidence rate of 0.53% among AGYW compared to 0.14% in young men [9,10]. The HIV prevalence in AGYW is 5.9%, which is almost double that of males, 3.3% [10]. About 87,000 AGYW were estimated to be living with HIV in 2019, of which 21,000 were undiagnosed [10]. Zimbabwe has made substantial progress towards the UNAIDS targets, having attained the last two targets, but it fell short of attaining the first target by 2020 [11]. The 2020 achievements were 87–97–90 for both males and females and 83–97–91 for females only [11]. Among AGYW, 76% of those living with HIV have been diagnosed [10,11]. This means that there is still a gap in HIV testing, which is a bottleneck towards achieving 2030 UNAIDS targets in Zimbabwe.

HIV testing is crucial in attaining the UNAIDS targets, because it is an entry point into HIV care. To achieve a reduction in new HIV infections through treatment as prevention, it is important to target key populations and vulnerable groups, such as AGYW [5]. Different studies have been done on HIV infection among AGYW in Africa, but there are few that have looked at the trends in self-reported HIV testing, especially among AGYW in Zimbabwe. The study analysed trends in self-reported HIV testing and evaluated factors associated with HIV testing uptake among AGYW in Zimbabwe using the latest publicly available Demographic and Health Surveys (DHS) data from 2005/6, 2010/11, and 2015. The study sought to identify some of the determinants of HIV testing uptake among AGYW, which can be considered in HIV programmes targeting this high-risk group.

## 2. Materials and Methods

### 2.1. Study Design

A secondary data analysis of DHS datasets from 2005/6, 2010/11, and 2015 was done. Cross sectional study designs were used in these surveys.

### 2.2. Study Setting

The Demographic and Health Surveys were conducted in all the districts of the 10 provinces of Zimbabwe. The samples were representative of the whole country having been drawn from both urban and rural areas.

### 2.3. Study Population and Sampling

The study population were AGYW who lived in Zimbabwe at the time of the surveys.

### 2.4. Inclusion Criteria

Female gender, aged between 15 and 24 years at the time of the survey, residents in Zimbabwe at time of survey and participated in the DHS for 2005/6, 2010/11, or 2015.

### 2.5. Exclusion Criteria

Eligible AGYW who did not answer questions in the section on HIV were excluded.

### 2.6. Sampling

In the surveys, stratified two-stage cluster random sampling of households were used. The Zimbabwe population census was used as a sampling frame. Each province is divided into districts, which are further divided into wards. For convenience, the wards are sub-divided into enumeration areas during a population census. The first stage used enumeration areas as the sampling units, whilst the second stage used households as the sampling units. Each survey had a household sample size of 10,800, 9756, and 10,534, respectively for the 2005/6, 2010/11, and 2015 surveys. All eligible members of the selected households who consented to participate were included in these surveys. Further details on sampling are available in the DHS manuals and the Zimbabwe DHS final reports for 2005/6, 2010/11, and 2015. The weighted sample sizes for AGYW in these surveys who fitted the inclusion criteria for this study were 4104, 3786, and 3895, respectively.

### 2.7. Measurements

Trained interviewers used structured questionnaires to collect data on many variables, including HIV knowledge, attitudes, and testing. Four different questionnaires were used for data collection, with one for households, one for biomarkers, one for women, and one for men. More information regarding the questionnaires is available from the DHS programme. A trend analysis was done using aggregated Zimbabwe DHS reports for 2005/6, 2010/11, and 2015.

A secondary data analysis of the Zimbabwe DHS 2015 available datasets was done on several variables using the women’s individual recode file to evaluate factors associated with HIV testing. The outcome variable was ever having an HIV test (yes or no). The independent variables included demographics such as age in five-year age groups (15–19 years or 20–24 years). Marital status had three responses (never married/in union, married/in union, formerly married/in union), where in union meant living together like a married couple.

Level of education had four responses (no education, primary, secondary, and higher than secondary), and residence was either rural or urban and wealth quintile/index (poorest, poorer, middle, richer, or richest). The wealth index is a composite measure of a household’s cumulative living standard calculated using data on a household’s ownership of selected assets. These include bicycles, televisions, sanitation facilities, water sources, and materials used for housing construction. A statistical procedure (principal components analysis) was used to generate the wealth index and placed individual households on a continuous scale of relative wealth. Employment in the last 12 months had multiple responses (currently, in the last six months, or no employment) and was recoded to either yes (currently or in the last six months) or no. All demographic variables had no missing data.

HIV knowledge included variables such as comprehensive HIV knowledge (yes or no), knowledge about MTCT (yes or no), and HIV non-discriminatory attitudes (yes or no), which were generated using responses from multiple questions, in which there were no missing data. Comprehensive HIV knowledge meant knowing that HIV infection can be reduced by having one uninfected faithful sexual partner and consistent condom use during sexual intercourse, that a healthy-looking individual can have HIV, that one cannot get HIV from mosquito bites, and that one cannot get HIV by witchcraft or supernatural means. Knowledge about MTCT meant knowing that HIV can be transmitted during pregnancy, delivery, or breastfeeding.

HIV non-discriminatory attitudes meant agreeing that children with and without HIV should be allowed to attend school together and that one would buy vegetables from an HIV-infected vendor. Knowing where to get an HIV test was a binary variable (yes or no), and missing data were excluded.

Risky sexual behaviour automatically excluded participants who never had sexual intercourse as well as missing responses; a ‘do not know’ response was treated as missing. The lifetime number of sexual partners was recoded from imputed numbers of one, two, and three or more. Condom use during last sexual intercourse and a history of receiving money or gifts in exchange for sex were binary variables (yes or no). The variable had an STI or STI symptoms in the past 12 months (yes or no) was generated using responses from multiple questions. This meant having any STI symptoms (genital sore/ulcer or genital discharge) or an STI diagnosis. The history of pregnancy was generated by recoding imputed age at first birth to yes (if age imputed) or no (if age not imputed). The variable for gender-based violence (GBV) was ever having experienced sexual violence, which was binary (yes or no).

### 2.8. Outcome Measures

The primary outcome measures of interest were whether one had ever been tested for HIV and whether one had received an HIV test in the past one year and received the test results.

### 2.9. Data Collection and Analysis

The study used a secondary data analysis of the most recent publicly available Zimbabwe DHS datasets from 2005/6, 2010/11, and 2015. The research was quantitative. Aggregated data from Zimbabwe DHS reports were used to determine trends in self-reported HIV testing. A trend analysis was used to analyse trends in HIV testing, and results were presented graphically. Descriptive statistics were used to summarise participant characteristics in the analysis of factors associated with self-reported HIV testing. The most recent DHS data from the 2015 survey was used to analyse the associated factors. The associations between self-reported HIV testing and independent variables were determined using bivariate and multivariable logistic regression models. Independent variables included age, marital status, level of education, residence, wealth index, employment, HIV knowledge, pregnancy, sexual behaviour, and GBV. A manual forward stepwise procedure was used to enter variables with *p*-values < 0.25 in the bivariate analysis into the multivariable model. Analyses were adjusted using survey weights, and a *p*-value < 0.05 was considered statistically significant. Odds ratios (OR) and their 95% confidence intervals were tabulated. Cases with missing data on either outcome or independent variable were excluded from the analysis. The statistical package used for analysis was STATA MP (version 14, StataCorp, College Station, TX, USA). Interpretation of the associations was used to evaluate whether the variables were associated with HIV testing uptake among AGYW.

## 3. Results

### 3.1. Overall Trends in HIV Testing

The proportions of AGYW who had HIV testing in aggregated DHS reports were used to analyse the trends in HIV testing. For the trend analysis among adolescent girls, aged 15–19 years, the proportion of those who ever had an HIV test increased from 14.7% in 2005/6 to 26.5% in 2010/11 and 47.9% in 2015 (Figure 1). For those who had an HIV test in the past 12 months and received results, the percentages increased from 4.8% in 2005/6 to 18.4% in 2010/11 and 29.8% in 2015 (Figure 2).

Among young women, aged 20–24 years, ever having an HIV test increased from 34.8% in 2005/6 to 68.7% in 2010/11 and 84.8% in 2015 (Figure 1). For those who had an HIV test in the past 12 months and received results, the proportions increased from 9% in 2005/6 to 42.2% in 2010/11 and 57.8% in 2015. Thus, there was an upward overall trend for both outcomes (Figure 2). The *p*-values (*p* < 0.001) for the increase observed in the trend analysis were statistically significant for both outcomes.

### 3.2. Analysis of Self-Reported HIV Testing Uptake in the 2015 Zimbabwe DHS

#### 3.2.1. Study Cohort

The 2015 DHS had a household sample size of 10,534, from which 9955 women, aged 15–49 years, were interviewed. Of these, 3938 AGYW participants, aged 15–24 years, who answered questions in the HIV/AIDS section of the questionnaire regardless of their sexual activity, were included in this study.

#### 3.2.2. Prevalence of Self-Reported HIV Testing

The prevalence of self-reported HIV testing (ever having an HIV test) among adolescent girls in this study was 47.9% (95% CI: 45.1 to 50.7), and among young women, it was 84.8% (95% CI: 82.5 to 86.8). The overall prevalence of self-reported HIV testing among AGYW was 63.9% (95% CI: 62.4 to 65.4).

#### 3.2.3. Demographic Characteristics of Study Participants

This study looked at the socio-demographic characteristics of AGYW as the study participants in the Zimbabwe DHS 2015. Among the AGYW, 56.4% were aged 15–19 years, and 43.6% were aged 20–24 years. Most participants were never married or in union (56.3%), whereas 37.9% were married or in union, and 5.8% were formerly married or in union. Only 0.3% of the participants had no education, 21.9% reached a primary education level, 74.2% reached a secondary education level, and 3.6% reached higher than a secondary education level. Most study participants resided in the rural areas (62.7%), versus 37.3% in the urban areas. About two-thirds (67.1%) of the study participants were unemployed in the past one year. For the wealth quintile category, the poorest had the least proportion of participants of 16.1%, while the richest had the highest proportion of 25.0% (Table 1).

#### 3.2.4. Socio-Demographic Factors Associated with Ever Having an HIV Test

Among the study participants, those aged 20–24 years had higher odds of testing for HIV compared to those aged 15–19 years (84.8% vs. 47.9%, OR 6.06, 95% CI: 5.05 to 7.26, *p* < 0.001). However, on adjusting for potential confounders, the odds of HIV testing in the 20–24 year-old participants were no longer significant (Table 2).

Among the study participants, those who were married or in union had higher odds of HIV testing than those who were never married or in union (87.9% vs. 45.0%, OR 8.86, 95% CI: 6.66 to 11.79, *p* < 0.001). After adjusting for potential confounders, this was no longer significant. Participants who were formerly married or in union had higher odds of HIV testing than those who were never married or in union (91.6% vs. 45.0%, OR 13.30, 95% CI: 8.00 to 22.09, *p* < 0.001). However, this was no longer significant after adjusting for potential confounders (Table 2).

Participants who lived in the rural areas had slightly higher odds of HIV testing than participants who lived in urban areas (65.8% vs. 60.9%, OR 1.23, 95% CI: 1.03 to 1.48, *p* = 0.021). This was no longer significant after adjusting for potential confounders. Participants who were employed in the past twelve months had higher odds of HIV testing than those who were not employed (73.7% vs. 59.2%, OR 1.93, 95% CI: 1.62 to 2.29, *p* < 0.001). However, this was no longer significant after adjusting for potential confounders.

The only statistically significant socio-demographic factor in the multivariable logistic regression model was education. Participants who reached a primary education level had higher odds of HIV testing than those with no education (63.8% vs. 39.3%, OR 2.72, 95% CI: 0.76 to 9.71, *p* = 0.122). This remained significant after adjusting for potential confounders (aOR 4.71, 95% CI: 1.23 to 18.02, *p* = 0.024). The odds of testing increased with an increasing level of education. Participants who reached a secondary education level had higher odds of HIV testing than those with no education (63.6% vs. 39.3%, OR 2.70, 95% CI: 0.76 to 9.57, *p* = 0.124). After adjusting for potential confounders, this remained significant (aOR 9.28, 95% CI: 2.42 to 35.48, *p* = 0.001). Participants who reached higher than a secondary education level had higher odds of HIV testing than those with no education (72.5% vs. 39.3%, OR 4.06, 95% CI: 1.05 to 15.69, *p* = 0.042). After adjusting for potential confounders, this remained significant (aOR 12.49, 95% CI: 2.69 to 57.92, *p* = 0.001) (Table 2).

#### 3.2.5. Risky Sexual Behaviour

Study participants who had two lifetime sexual partners had higher odds of HIV testing than those who had one lifetime sexual partner (91.1% vs. 84.4%, OR 1.89, 95% CI: 1.20 to 2.98, *p* = 0.003). This was no longer significant after adjusting for potential confounders. Participants who had three or more lifetime sexual partners had higher odds of HIV testing than those who had one lifetime sexual partner (92.5% vs. 84.4%, OR 2.28, 95% CI: 1.31 to 3.94, *p* = 0.003). This remained significant after adjusting for potential confounders (aOR 2.0, 95% CI: 1.09 to 3.66, *p* = 0.025). Participants who used condoms during last sexual intercourse had slightly lower odds of HIV testing than those who used no condoms (85.6% vs. 86.6%, OR 0.92, 95% CI: 0.63 to 1.35, *p* = 0.669), but this was statistically not significant. Participants who received money or gifts in exchange for sex had lower odds of HIV testing than those who did not (78.0% vs. 82.1%, OR 0.77, 95% CI: 0.31 to 1.94, *p* = 0.583), but this was not statistically significant. Participants who had an STI or STI symptoms had lower odds of HIV testing than those who did not (80.4% vs. 87.0%, OR 0.61, 95% CI: 0.41 to 0.92, *p* = 0.018) (Table 2).

#### 3.2.6. Knowledge about HIV

Study participants with comprehensive HIV knowledge had higher odds of HIV testing than those without (69.7% vs. 58.9%, OR 1.60, 95% CI: 1.35 to 1.89, *p* < 0.001). After adjusting for potential confounders, the odds of HIV testing remained significant (aOR 1.91, 95% CI: 1.31 to 2.78, *p* = 0.001). Participants who knew about mother-to-child transmission (MTCT) of HIV had higher odds of testing than those who did not (71.6% vs. 52.2%, OR 2.31, 95% CI: 1.99 to 2.68, *p* < 0.001). This remained significant after adjusting for potential confounders (aOR 2.09, 95% CI: 1.55 to 2.82, *p* < 0.001). Participants who had non-discriminatory attitudes towards people living with HIV had higher odds of testing than those who had discriminatory attitudes (68.0% vs. 54.7%, OR 1.76, 95% CI: 1.48 to 2.11, *p* < 0.001). This remained significant after adjusting for potential confounders (aOR 1.60, 95% CI: 1.12 to 2.28, *p* = 0.010) (Table 2).

#### 3.2.7. Pregnancy and GBV

Study participants who had a history of pregnancy had higher odds of HIV testing than those without (92.5% vs. 44.8%, OR 15.18, 95% CI: 11.12 to 20.72, *p* < 0.001). This remained significant after adjusting for potential confounders (aOR 6.08, 95% CI: 4.22 to 8.75, *p* < 0.001). Participants who had a history of sexual violence had higher odds of HIV testing than those without (78.0% vs. 68.9%, OR 1.60, 95% CI: 1.16 to 2.21, *p* = 0.005). This was no longer significant after adjusting for potential confounders (Table 2).

## 4. Discussion

The study analysed trends in self-reported HIV testing from 2005 to 2015 and evaluated factors associated with self-reported HIV testing from the 2015 DHS among AGYW in Zimbabwe. These are important, as Zimbabwe is working towards achieving UNAIDS 95–95–95 goals by 2030.

The study showed an overall low uptake of HIV testing across all the surveys (2005/6, 2010/11, and 2015) and in both age groups (15–19 and 20–24 years). The proportion of participants who had ever received an HIV test was higher in young women, aged 20–24 years, than in adolescent girls, aged 15–19 years. The same trend was shown for those who had an HIV test in the past one year and received results. However, the proportion of participants who were tested for HIV in the past one year was lower than for those who were ever tested for HIV. Only 29.8% of adolescents had an HIV test in the past one year and received results in 2015 compared to 47.9% of those who ever had an HIV test. The same trend was seen for young women: 57.8% had an HIV test in the past one year compared to 84.8% who had ever received an HIV test. This reflects the low frequency of HIV testing among AGYW, which is concerning given the high incidence of HIV infection in this age group. Therefore, programs targeting AGYW in Zimbabwe should make them aware of the continuous risk of HIV acquisition and encourage frequent HIV testing for early diagnosis and treatment.

Other studies have also reported low HIV testing rates among AGYW [12,13,14,15]. In a WHO research project (2013), they found that only 65% of AGYW had ever been tested for HIV [15]. Similarly, Giguere et al. (2021) found that 65% of AGYW in SSA knew their HIV status in 2020 [16]. In a South African study, Madiba et al. (2015) found that only half of the study participants had an HIV test in the past one year [12]. Several factors contribute to a low HIV testing uptake among AGYW. The unavailability of youth-friendly HTC services and inaccessible healthcare facilities contribute to a low HIV testing uptake [12]. Healthcare workers’, community’s, and adolescent’s attitudes towards HTC in this age group might contribute to a low HIV testing uptake in Zimbabwe. This is because HIV is perceived to be associated with sexual activity, and adolescents are expected to abstain from sex.

In this study, there was an increase in self-reported HIV testing from 2005 to 2015. A South African study by Jooste et al. (2020) also showed an increase in self-reported HIV testing among young people from 18.3% in 2005 to 58.8% in 2017 [17]. Similarly, a Zambian study by Heri et al. (2021) showed an increase in HIV testing over time among non-pregnant AGYW from 10% in 2007 to 58% in 2018 [18]. The increase in HIV testing from 2005 to 2015 in Zimbabwe might be attributed to an increased awareness of HIV among AGYW and different interventions, such as provider-initiated HTC offered in healthcare facilities. Other studies have shown higher HIV testing rates among young women than adolescent girls [12,13,19,20]. This could be due to higher HIV awareness in the older age group and that they are more likely to have been pregnant and attended ANC where HIV testing is offered routinely. The age of consent in Zimbabwe is 16 years, and this might contribute to adolescent girls below 16 years not getting tested for HIV.

The study found that several factors were associated with self-reported HIV testing. Increasing the level of education, having more than three lifetime sexual partners, comprehensive knowledge of HIV, knowledge about MTCT, HIV non-discriminatory attitudes, and a history of pregnancy were all significantly associated with self-reported HIV testing.

The association between level of education and self-reported HIV testing is like other studies [17,18,20,21]. This might be due to increasing knowledge and understanding about HIV with an increasing level of education, as more is taught in schools. However, despite the fact that Zimbabwe has a high literacy rate, this has not translated into positive HIV testing attitudes, given the low uptake found in the trend analysis.

The study found an association between three or more lifetime sexual partners and self-reported HIV testing. Other studies have also found this association with an increasing number of lifetime sexual partners [18,20,21]. In these studies, the odds of HIV testing were highest in those with five or more sexual partners [20,21]. This is probably because those who have multiple sexual partners perceive themselves to be at an increased risk of getting HIV infection and, therefore, seek HTC services more. However, in a study by Madiba et al. (2015), no association was found between the number of sexual partners and HIV testing [12]. This may be because the study was done in high schools, where the participants were mostly adolescents who are younger, whereas in this study, the sample was nationally representative of AGYW, in which young women who are older and more educated were included.

The study showed that comprehensive knowledge of HIV, knowledge about MTCT of HIV, and non-discriminatory attitudes towards PLWHIV were significantly associated with HIV testing. This agreed with other literature [17,18,19,20]. The association is probably because those who know more about HIV are more likely to want to know their status. Furthermore, those who do not discriminate between HIV positive individuals are more likely to go for HIV testing, since they are more accommodating.

In addition to the factors already mentioned, pregnancy was strongly associated with HIV testing. Other studies documented this association [19,20,21]. This finding may be because most pregnant women who attend ANC in Zimbabwe are offered HIV testing. In the current study, 92.5% of all those with a history of pregnancy were tested for HIV, reflecting the high rate of acceptance during ANC and the success of prevention of mother-to-child transmission (PMTCT) of HIV programmes in Zimbabwe. This is particularly important for the elimination of paediatric HIV infections [22].

The study findings showed several important factors that were not associated with HIV testing. Marital status was not associated with HIV testing after adjusting for potential confounders. Similarly, Heri et al. (2021) found no significant association between those currently in union and HIV testing among non-pregnant AGYW in Zambia [18]. Contrary to our study, Jooste et al. (2020) and Takarinda et al. (2016) found associations between being married or formerly married and HIV testing [17,20]. This is probably because the study populations included both youths and older adults who are more likely to have been tested, particularly women of reproductive age, during ANC. In addition, more opportunities for HIV testing are probably encountered by older married women than AGYW. Considering that HIV transmission in Zimbabwe is largely through heterosexual intercourse, married or formerly married AGYW should be targeted for HIV testing. Additionally, most married AGYW are probably in age-disparate relationships. The older male partners have higher chances of being HIV infected, thereby increasing the risk of HIV transmission to the younger partners [23]. Thus, AGYW have a high incidence of HIV, especially in SSA [3].

The current study did not find a significant association between accepting money or gifts in exchange for sex and HIV testing. Similarly, Musekiwa et al. (2020) found no association [19]. This is probably because AGYW might not consider the risk of HIV infection associated with such behaviour. In most cases, the male partners paying for sex are probably older, thereby increasing the risk of HIV infection. Therefore, health promotion targeting behaviour change and positive attitudes towards HIV testing among AGYW might reduce this gap between risky sexual behaviour and the uptake of HIV testing.

Having an STI or STI symptoms was not associated with HIV testing in this study, despite the fact that these increase the risk of HIV infection. This reflects infrequent visits to healthcare facilities by AGYW or a lack of integration within different programmes, such as HTC services, sexual and reproductive health, and general consultations. Heri et al. (2021) found similar results [18]. Similarly, in a study by Takarinda et al., there was no significant association between having an STI or STI symptoms and HIV testing [20].

This study found no association between a history of sexual violence and ever having an HIV test. Musekiwa et al. (2020) obtained similar results in a study done in South Africa [19]. This may be due to a lack of women empowerment and societal norms, especially among AGYW, who may not make their own decisions [3].

The factors discussed above were not significantly associated with ever having an HIV test in this study, despite the fact that they increase the risk for HIV acquisition. Therefore, they are important factors to consider when planning interventions aimed at improving HIV testing uptake among AGYW. The lack of association in the current study probably reflects negative behaviours and attitudes towards HIV testing among AGYW, which should be addressed.

## 5. Conclusions

The study showed an overall increase in HIV testing among AGYW in Zimbabwe from 2005 to 2015, which was much more in young women than adolescent girls. However, HIV testing was low considering the UNAIDS 95–95–95 goals to be achieved by 2030. The factors associated with self-reported HIV testing included education, an increasing number of lifetime sexual partners, comprehensive knowledge of HIV, knowledge about MTCT of HIV, non-discriminatory attitudes towards PLWHIV, and a history of pregnancy. Marital status, ever receiving money or gifts in exchange for sex, a history of STI or STI symptoms, and history of sexual violence showed no significant association with HIV testing, despite the fact that they all increase the risk for HIV transmission. Therefore, these factors should be addressed within interventions targeting AGYW. There is a need to promote and sustain education of the female child and strengthen HIV awareness and behaviour change among AGYW to improve HIV testing uptake. There is a need to scale-up national HIV prevention programmes targeting AGYW, especially adolescent girls, aged 15–19 years. Further research among AGYW is needed that considers how challenges caused by the COVID-19 pandemic may have impacted sexual and reproductive health and other services, including HIV testing.

## Figures and Tables

**Figure 1 ijerph-19-05165-f001:**
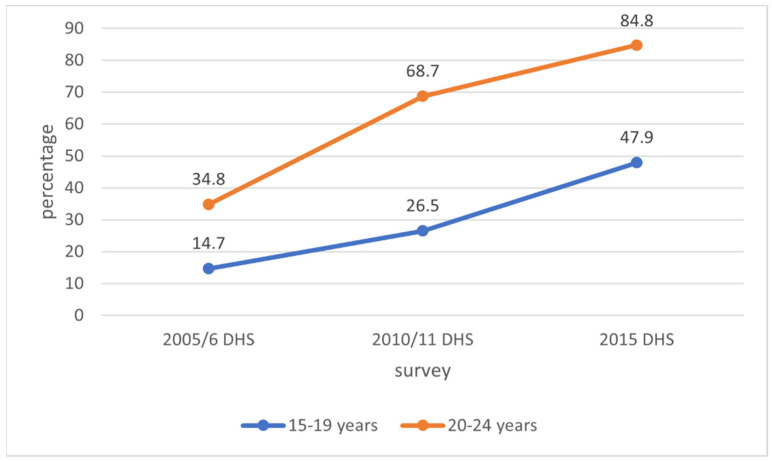
Trends in ever having an HIV test among adolescent girls and young women in Zimbabwe, 2005 to 2015.

**Figure 2 ijerph-19-05165-f002:**
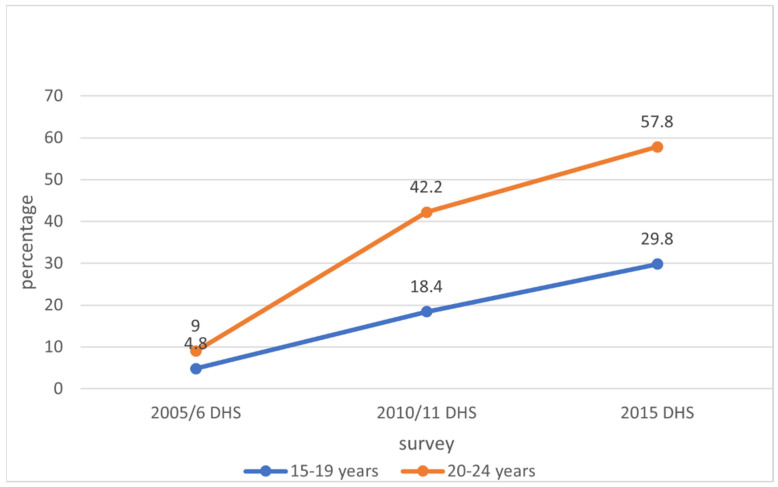
Trends in having an HIV test in the past one year and receiving results among adolescent girls and young women in Zimbabwe, 2005 to 2015.

**Table 1 ijerph-19-05165-t001:** Socio-demographic characteristics of AGYW in Zimbabwe DHS 2015.

Characteristic	Category	*n*	Percent ^a^	95% CI
Age in years	15–19	2156	56.4	54.6 to 58.3
	20–24	1782	43.6	41.7 to 45.4
Marital status	Never married/in union	2279	56.3	54.0 to 58.6
	Married/in union	1440	37.9	35.8 to 40.1
	Formerly married/in union	219	5.8	5.0 to 6.7
Education	No education	11	0.3	0.2 to 0.5
	Primary	801	21.9	19.7 to 24.2
	Secondary	2967	74.2	71.9 to 76.4
	Higher	159	3.6	2.9 to 4.6
Residence	Urban	1744	37.3	34.6 to 40.0
	Rural	2194	62.7	60.0 to 65.4
Employment in the last 12 months	Yes	1270	32.9	30.9 to 35.0
	No	2668	67.1	65.0 to 69.1
Wealth quintile	Poorest	573	16.1	14.0 to 18.4
	Poorer	607	17.9	16.2 to 19.8
	Middle	644	18.6	16.6 to 20.7
	Richer	997	22.4	19.7 to 25.3
	Richest	1117	25	21.9 to 28.5

^a^—weighted percentage; CI—confidence intervals; HIV—Human Immunodeficiency Virus.

**Table 2 ijerph-19-05165-t002:** Factors associated with HIV testing among adolescent girls and young women who participated in the 2015 Zimbabwe DHS.

				Univariate			Multivariate		
Variable	Category	*n*	% HIV Tested ^a^	OR	95% CI	*p*-Value	aOR	95% CI	*p*-Value
**Socio-Demographic Characteristics (*n* = 3938)**									
Age in years	15–19	2156	47.9	Ref					
	20–24	1782	84.8	6.06	5.05 to 7.26	<0.001	1.14	0.83 to 1.56	0.422
Marital status	Never married/in union	2279	45	Ref					
	Married/in union	1440	87.9	8.86	6.66 to 11.79	<0.001	1.29	0.87 to 1.93	0.199
	Formerly married/in union	219	91.6	13.3	8.00 to 22.09	<0.001	1.31	0.69 to 2.49	0.401
Residence	Urban	1744	60.9	Ref					
	Rural	2194	65.8	1.23	1.03 to 1.48	0.021	1.24	1.01 to 1.52	0.04
Education	No education	11	39.3	Ref					
	Primary	801	63.8	2.72	0.76 to 9.71	0.122	**4.71**	**1.23 to 18.02**	**0.024**
	Secondary	2967	63.6	2.7	0.76 to 9.57	0.124	**9.28**	**2.42 to 35.48**	**0.001**
	Higher	159	72.5	4.06	1.05 to 15.69	0.042	**12.49**	**2.69 to 57.92**	**0.001**
Employment	No	2668	59.2	Ref					
	Yes	1270	73.7	1.93	1.62 to 2.29	<0.001	1.22	0.98 to 1.50	0.066
Wealth quintile	Poorest	573	67	Ref					
	Poorer	607	66.6	0.98	0.74 to 1.29	0.886			
	Middle	644	65.6	0.94	0.72 to 1.23	0.646			
	Richer	997	65	0.91	0.69 to 1.21	0.534			
	Richest	1117	57.8	0.67	0.52 to 0.88	0.004			
**Risky sexual behaviour**									
Lifetime number of sexual partners (*n* = 2210)	1	1527	84.4	Ref					
	2	413	91.1	1.89	1.20 to 2.98	0.006	1.61	0.99 to 2.60	0.055
	3 or more	270	92.5	2.28	1.31 to 3.94	0.003	**2**	**1.09 to 3.66**	**0.025**
Condom use at last sexual intercourse (*n* = 1988)	No	1605	86.6	Ref					
	Yes	383	85.6	0.92	0.63 to 1.35	0.669			
Ever received money/gifts in exchange for sex (*n* = 573)	No	549	82.1	Ref					
	Yes	24	78	0.77	0.31 to 1.94	0.583			
Had an STI	No	2008	87	Ref					
	Yes	205	80.4	0.61	0.41 to 0.92	0.018	0.66	0.4 to 1.09	0.106
**Knowledge about HIV**									
Comprehensive HIV knowledge ^b^	No	2059	58.9	Ref					
	Yes	1879	69.7	1.6	1.35 to 1.89	<0.001	**1.91**	**1.31 to 2.78**	**0.001**
Knowledge of MTCT ^c^	No	1498	52.2	Ref					
	Yes	2440	71.6	2.31	1.99 to 2.68	<0.001	**2.09**	**1.55 to 2.82**	**<0.001**
Non-discriminatory attitudes ^d^	No	1165	54.7	Ref					
	Yes	2773	68	1.76	1.48 to 2.11	<0.001	**1.6**	**1.12 to 2.28**	**0.01**
**Pregnancy**									
Ever been pregnant	No	2342	44.8	Ref					
	Yes	1596	92.5	15.18	11.12 to 20.72	<0.001	**6.08**	**4.22 to 8.75**	**<0.001**
**GBV**									
Ever experienced sexual violence	No	2198	68.9	Ref					
	Yes	279	78	1.6	1.16 to 2.21	0.005			

%—percent; HIV—Human Immunodeficiency Virus; OR—odds ratio; aOR—adjusted odds ratio; CI—confidence interval; GBV—gender based violence; Ref—reference category; MTCT—mother-to-child transmission of HIV, ^a^—weighted percentages; ^b^—Comprehensive HIV knowledge meant knowing that HIV infection can be reduced by having one uninfected faithful sexual partner and consistent condom use during sexual intercourse, that a healthy looking individual can have HIV, that one cannot get HIV from mosquito bites, and that one cannot get HIV by witchcraft or supernatural means; ^c^—HIV knowledge about MTCT meant knowing that HIV can be transmitted during pregnancy, delivery, or breastfeeding; ^d^—Non-discriminatory attitudes meant agreeing that children with and without HIV should be allowed to attend school together and that one would buy vegetables from an HIV infected vendor; Figures in bold are statistically significant.

## Data Availability

Publicly available datasets were analysed in this study. This data can be found here: https://dhsprogram.com/data/available-datasets.cfm (accessed on 16 March 2021).

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
