# Peer review of "Trends in HIV Testing and Associated Factors among Adolescent Girls and Young Women in Zimbabwe: Cross-Sectional Analysis of Demographic and Health Survey Data from 2005 to 2015"

_ijerph, 2022, doi:10.3390/ijerph19095165_

Round 1

Reviewer 1 Report

The authors have studied HIV data in adolescent and young women in Zimbabwe and analyzed correlation of various demographic factors affecting the increase in self reporting. This kind of analysis is valuable to understand the trends and the factors affecting it so that appropriate action can be taken. The approach and methods, although simple and routine, have been presented professionally and comprehensively. The authors also present the discussion with a clear message of what led to the increase in numbers and what the recommendations are to meet 95-95-95 goal. Overall, the paper is well written and presented. The paper could be improved with additional visual depiction of data and highlighting results that are statistically significant.  One minor comment: the quality of graphs can be improved to a professional standard. The paper is acceptable to me in the current form.

Author Response

Thank you for your comments. The results that are statistically significant have been highlighted in bold in table 2. The resolution of the graphs have been increased to 300 dpi as specified in the Authors instruction to authors. 

Reviewer 2 Report

The manuscript by Pachena and Musekiwa analyzed the trends in HIV testing among young women in Zimbabwe, a country with a high prevalence of HIV infection, to identify possible variables affecting the tendency to test for HIV infection. 

Although not particularly novel the study is well designed and conducted;  results are well described, and somewhat "expected". 

My only major concern regards the statistical analysis: in the material and methods section, the statistical model and statistical tests used are not described in detail, so it is hard to make any judgment on the robustness of the analysis performed. This section must be improved and clarified. 

Author Response

Thank you for your comments. More detail was given for the multivariable logistic regression model used during statistical data analysis.

Reviewer 3 Report

The manuscript by Pachena and Musekiwa reports the results of a cross-sectional analysis of the incidence of HIV in adolescent girls and young women, aged 15-245 years, from sub-Saharan Africa. 

The analysis is methodical and the results are clearly presented in the manuscript. This data is critical in the policy decisions of various international HIV control programs. 

Author Response

Thank you for your comments.

Round 2

Reviewer 2 Report

The authors have replied my concerns in a satisfactory manner.